# Invasive Urban Mammalian Predators: Distribution and Multi-Scale Habitat Selection

**DOI:** 10.3390/biology11101527

**Published:** 2022-10-19

**Authors:** Kim F. Miller, Deborah J. Wilson, Stephen Hartley, John G. Innes, Neil B. Fitzgerald, Poppy Miller, Yolanda van Heezik

**Affiliations:** 1Department of Zoology, University of Otago, P.O. Box 56, Dunedin 9054, New Zealand; 2Manaaki Whenua—Landcare Research, Private Bag 1930, Dunedin 9054, New Zealand; 3Centre for Biodiversity and Restoration Ecology, School of Biological Sciences, Victoria University of Wellington, P.O. Box 600, Wellington 6140, New Zealand; 4Manaaki Whenua—Landcare Research, Private Bag 3127, Hamilton 3240, New Zealand; 5Plant & Food Research, 23 Batchelar Road, Palmerston North 4410, New Zealand

**Keywords:** hedgehog, brushtail possum, house mouse, rat, city, residential garden, *Mus musculus*, *Rattus norvegicus*, *Rattus rattus*, *Erinaceus europaeus*, *Trichosurus vulpecula*

## Abstract

**Simple Summary:**

Restoration of biodiversity in urban green spaces frequently requires eradication or management of invasive species. We aimed to identify fine- and landscape-scale habitat features associated with the presence of five invasive urban mammals (*Rattus* species, European hedgehogs, mice, and brushtail possums) in three urban green space types (forest fragment, amenity park, residential garden) across three New Zealand cities, and across two seasons, to identify where management effort should be focused. All species were detected in all greenspace types; however, rodents were detected least in residential gardens, possums were detected most often in forest fragments, and hedgehogs least in forest fragments. Proximity of amenity parks to forest patches was positively associated with possum and hedgehog presence and negatively with rats. Conversely, proximity of residential gardens to forest patches was positively associated with rat presence. Management of rats should focus on sites with shrub and lower canopy cover and of mice on sites with herb layer cover, while micro-habitat features were not important for hedgehogs and possums. Rats were most likely to be found in residential gardens with compost heaps. The wide distributions of these species suggest that in order to be successful, ecological restoration must be coordinated, target all green space types, and engage urban residents.

**Abstract:**

A barrier to successful ecological restoration of urban green spaces in many cities is invasive mammalian predators. We determined the fine- and landscape-scale habitat characteristics associated with the presence of five urban predators (black and brown rats, European hedgehogs, house mice, and brushtail possums) in three New Zealand cities, in spring and autumn, in three green space types: forest fragments, amenity parks, and residential gardens. Season contributed to variations in detections for all five taxa. Rodents were detected least in residential gardens; mice were detected more often in amenity parks. Hedgehogs were detected least in forest fragments. Possums were detected most often in forest fragments and least often in residential gardens. Some of this variation was explained by our models. Proximity of amenity parks to forest patches was strongly associated with presence of possums (positively), hedgehogs (positively), and rats (negatively). Conversely, proximity of residential gardens to forest patches was positively associated with rat presence. Rats were associated with shrub and lower canopy cover and mice with herb layer cover. In residential gardens, rat detection was associated with compost heaps. Successful restoration of biodiversity in these cities needs extensive, coordinated predator control programmes that engage urban residents.

## 1. Introduction

Despite large-scale habitat loss brought about by urban expansion [1,2], cities still provide habitat for many species, both native and introduced [3,4,5], and even refuges for some endangered species [6]. Urban green spaces are most frequently the sites that provide such habitat [4,6,7,8]. The variety of green space types, from remnant patches of native vegetation, to highly artificial and engineered green infrastructure, such as green roofs [9,10], results in considerable variation across green spaces in terms of the biodiversity they support [11]. Size, connectivity, vegetation composition and structure of green spaces all affect their capacity to sustain biodiversity [12,13,14]. Nevertheless, despite the often small size of habitat patches and the additional stressors of human disturbances and introduced predators, green spaces can support a diversity of native and introduced fauna, and play a valuable role in sustaining biodiversity [6,14].

Recognition of the various benefits that urban green spaces provide, both for biodiversity and for human well-being, has resulted in a resurgence of initiatives to restore the quality of these spaces [15,16,17,18]. However, there are numerous challenges when undertaking ecological restoration, especially in an urban context [19,20,21]. In New Zealand (NZ) one of the key barriers to success is invasive species [22,23], particularly invasive predatory mammals, which comprise one of the foremost drivers of species’ declines in NZ and globally [24,25]. When humans first arrived, NZ ecosystems contained no land mammals other than some species of bat [2]. Currently there are 12 predatory mammals including four species of rodent (*Rattus* spp. & *Mus musculus*), three mustelids, European hedgehogs (*Erinaceus europaeus*), common brushtail possums (*Trichosurus vulpecula*), pigs (*Sus scrofa*), cats (*Felis catus*) and dogs (*Canis lupus familiaris*). Of these, all but pigs, *Rattus exulans*, and mustelids are commonly found in urban areas. Because urban green spaces provide habitat that supports populations of these invasive predators, their control or eradication from these areas will contribute to the success of urban restoration projects. 

Brown rats (*Rattus norvegicus*) are significant predators of ground-nesting and low-nesting bird species, large ground-dwelling invertebrates [26,27], and lizards [28], and they negatively affect vegetation recruitment [29]. They are also the dominant rat in most cities worldwide [30]. Black rats (*Rattus rattus*) are more common than brown rats in NZ urban environments [31]; their exceptional climbing abilities have enabled them to easily exploit nesting birds, and they have been implicated in the decline and extinction of many NZ bird species [32]. While primarily an insectivore, hedgehogs are also known as predators of lizards, ground-nesting birds and native snails [33,34]. The house mouse has an omnivorous diet and is a significant predator of seeds [35,36], invertebrates [37,38], and herpetofauna [39,40], and may compete for food with native species [41]. Brushtail possums are destructive browsers on native vegetation, competing with native birds for food and nesting resources [42,43], and preying on eggs, nestlings and adult birds [44,45].

As efforts to restore urban green space gain momentum, the need to understand urban invasive predator ecology becomes increasingly important, with a better understanding of habitat preferences being a key step towards the successful control of mammalian invasive predators across the range of urban green space types. Managing animal populations requires information on where animals are, why they are there, and where else they could be [46]. Multi-scale habitat selection studies provide insights into resource selection at micro- and landscape scales [47,48], addressing the “why”, and improving our understanding of potential drivers of animal distributions [47].

Broad-scale habitat selection studies on medium and small-sized predatory mammals have shown that urban environments provide adequate resources for several generalist species. For example, brown rats select habitats that provide adequate shelter, food and water [49,50]. Red foxes (*Vulpes vulpes*) in the UK and Australia find ideal habitat in the form of established gardens with hedges and shrubs that provide cover, and parks and reserves with thickets of non-native plants that provide diurnal nest sites [51,52,53]. In the US, coyotes (*Canis latrans*) avoid land cover types associated with human activity, such as residential areas, managed lawns and parks and commercial and industrial areas, and are mostly found in patches of natural habitat [54]. Fine-scale habitat selection studies have been applied more frequently to small mammals: in non-urban environments there are strong associations with habitat features such as grass, shrub and canopy cover, vertical stem density, debris and distance to trees( e.g., [55,56]). In urban areas, vegetation density and structure, availability of nesting habitat, and management intensity all influence distributions and densities of small mammals, with frequently tended, manicured green spaces such as parks and gardens supporting fewer small mammals [57,58]. 

In New Zealand most habitat selection studies on small and medium-sized introduced mammals have been done in wild or rural environments [59,60,61,62,63]. A small number of urban studies have revealed some habitat associations: brushtail possums, mice and rats tend to be found in forest fragments mostly, but also in more modified habitats such as private gardens and parks. Hedgehogs are abundant in a range of habitats [31,64,65,66] that have a variety of structures such as bushes, trees, heaps of branches and stones, which provide secure resting, breeding and hibernating sites [13,67,68]; they often prefer garden habitat [13,69]. Most studies to date have focused on one or two species in a specific habitat type or city. Broad-scale, multi-habitat and multi-species research which allows us to understand factors influencing distributions of urban invasive predatory species across a range of urban green space types and in different urban centres should better inform efforts to limit negative impacts on native biodiversity.

We aimed in this study to (1) identify common features determining urban predator distributions by conducting our research in three cities (Dunedin, Wellington, and Hamilton), and in three types of urban green space (forest fragment, amenity green space and residential gardens); and (2) improve our understanding of the habitat characteristics driving distributions of invasive mammalian predators across green spaces to inform the design of urban restoration projects and urban pest control. Forest fragments are important sources of native diversity as they primarily consist of native flora, have large trees, and often dense vegetation, all of which are associated with higher levels of biodiversity [70,71,72]. Urban parks, such as walking parks and sports fields, are areas of relatively open green space usually dominated by large grassy areas with patches of trees and shrubs and water features [73]. Although structurally very different, and individually much smaller, residential gardens cumulatively make up the largest proportion of urban green space across many cities, supporting biodiversity and providing ecosystem services [74,75,76]. Gardens, which are highly variable in size, structural composition, and species diversity [75,76] tend to be dominated by exotic species, with few large trees [72,77]. Widespread adoption of wildlife-friendly gardening activities, including control of invasive predators, has the potential to substantially enhance the benefits that gardens contribute to city-wide biodiversity and public health [12,72,76]. 

We determine the fine-scale and landscape-scale habitat characteristics that influence the distributions of the following five non-companion, mammalian, invasive predators found in urban green spaces: black rats, brown rats, hedgehogs, house mice, brushtail possums. These species are known to occur in all NZ cities and can be controlled using widely applied methods (e.g., traps). A better understanding of the distribution of these species across the city and their fine-scale habitat associations will allow more efficient control, such as more targeted trap placements. We do not examine the habitat preferences of mustelids, which were detected infrequently and are known to be uncommon in cities [31], nor of dogs and cats, as their companion animal status renders their control more complex [78], despite, in the case of cats, being important urban predators [79]. However, we do report on the prevalence of these species.

Our final objective was to test the broader generality of our species-specific habitat association models by comparing them to models fitted to similar data collected from two additional cities.

## 2. Materials and Methods

### 2.1. Study Design and Detection Devices

Surveys of mammalian predators were carried out in three New Zealand cities: Dunedin (91.6 km^2^), Wellington (112.4 km^2^), and Hamilton (110.4 km^2^; Figure 1) [80]. Three types of green space were sampled: (i) forest fragments (patches of primary/remnant and secondary (i.e., planted > 10 years ago) predominantly native forest), (ii) amenity parks, and (iii) residential gardens. Amenity parks were highly modified areas (parks/reserves/sports fields), which were fringed with both native and introduced scattered trees, shrubs, and/or long grass. Residential properties had significant garden vegetation present (i.e., paved yards and gardens with minimal vegetation were not selected for study).

Data were collected during two sampling seasons, early November to mid-December (spring 2017), and late April to mid-June (autumn 2018). In Dunedin and Hamilton there were 12 transect lines, four in each habitat type, and in Wellington there were 24, eight per habitat type. Transects were mostly situated a minimum of 500 m apart. In forest patches and amenity parks transects were approximately 450 m long, with 10 sampling stations spaced 50 m apart. Where necessary, transect lines were bent to fit into habitat boundaries. In residential gardens, station spacing was irregular and lines ranged in length between 300 m and 650 m.

At each station we deployed two detection devices 4–10 m apart: a footprint tracking tunnel and a chew card. Tracking tunnels were Black Trakka™ (100 mm × 100 mm × 500 mm), with Black Trakka™ inked tracking cards. Chew cards (90 mm × 180 mm × 3 mm) were pre-baited with peanut butter-flavoured possum dough (Traps.co.nz, Christchurch) and nailed to trees 30 cm above the ground or attached to wire pegs 15 cm above the ground. Tracking tunnels baited with a food lure were placed in locations with close cover where possible, i.e., under bushes or next to trees or logs, and were held down with wire pegs. 

Motion-activated infra-red cameras were placed facing the tracking tunnel at two of the ten stations on each line, situated 100–400 m apart. Cameras (Bushnell Trophy Cam Aggressor and some Reconyx 500) were set 50 cm above the ground, aimed at the tracking tunnel, which was placed broadside 150 cm away (at >200 cm distance the detection of rats declines) [81]. Cameras were active 24 h per day and took three still photographs (8 megapixels) each time they were triggered. The minimum time between triggers was 30 s, to avoid collecting an excessive number of photos of the same individual, and to save on memory card space. Image format was set to full screen, and LED control to medium. Sensitivity level was set to high to incur the highest chance of detecting small mammals such as mice. The night vision (NV) shutter speed was set to medium. A slower shutter speed permits more light to hit the sensor, increasing visibility but reducing picture quality when there is movement. The medium setting provides a compromise between visibility and blurring.

In each season, devices were deployed for a one-night exposure with peanut butter used as a lure in the tracking tunnel. Immediately following this and over six nights, tracking cards and chew cards were replaced and tunnels were baited with Erayz paste (a non-toxic rabbit-based bait, Connovation Ltd., Auckland, New Zealand), designed to attract mustelids, which have large home ranges and are difficult to detect in a single night [82,83].

### 2.2. Identification of Predators

Species were identified from ink footprints and teeth impressions; identifications from Dunedin, Wellington and Hamilton were cross-checked by two researchers. Because footprints and chew marks from rats and mustelids cannot reliably be identified to species level, they were recorded simply as rat or mustelid. Photos from cameras were identified to the species level where possible. 

### 2.3. Habitat Surveys

Habitat surveys were conducted at each station in a 10 m diameter circle centered on the position of the tracking tunnel, using a modified vegetation RECCE plot format [84]. Vegetation was stratified into six height tiers: herb layer (vegetation < 30 cm high), shrub layer (0.3–2 m high), sub-canopy (2–5 m), lower canopy (5–12 m), mid canopy (12–25 m), and high canopy (>25 m). Within each height tier, the total vegetation cover was scored categorically from 0–6, corresponding to 0%, 1%, 1–5%, 6–25%, 26–50%, 51–75%, and 76–100% cover, where cover was defined as the area of ground covered from a bird’s eye view; hence, stems and trunks were considered cover. Cover of leaf litter, rock, bare soil, woody debris (>1 cm diameter), low artificial cover (e.g., concrete, benches, decking), and cover from buildings were estimated using the same classes. Diameter at breast height (DBH) of the largest tree within each surveyed circle was also measured. 

At residential garden stations, an additional set of habitat variables were assessed within a 20 × 20 m quadrat centered on the tracking tunnel, or across the whole backyard (whichever was smaller). These additional variables were proportion of vegetation cover that was native, and proportions of ground covered by artificial hard landscaping features (i.e., buildings, decking, paving, fences; differing from the earlier measures owing to the greater area assessed), by mown lawns, and by regularly turned soil (flower and vegetable beds, etc.). Scores were cover classes 1–6, corresponding to <10%, 10–25%, 26–50%, 51–75%, 76–90%, and >90% cover. Two further categorical variables were recorded: compost heaps (present or absent), and level of property maintenance (low, medium, high). Property maintenance was evaluated based on apparent regular lawn mowing, maintained hedges, and evidence of pruning and weeding. 

Landscape-scale attributes were measured at each station with geographical information system software (qGIS v 3.8.3). These were distance (Euclidean) to coast, distance to above-ground freshwater bodies, distance to the outer edge of patches of forest larger than 1 ha, and distance to open grassland/pasture larger than 1 ha. 

### 2.4. Data Transformation

At each station, the four detection opportunities from the chew cards and tracking tunnels over the 1-night and 6-night periods were aggregated to a single record of presence/absence over the entire 7-night period. At each station with a camera, an additional presence/absence event was recorded for the same period. A categorical predictor variable indicating the detection method enabled comparison between methods (tracking tunnel and chew card vs camera; see below).

Cover scores were converted to the mid-point percent cover of each category range i.e., RECCE cover scores 0–6 were converted to 0%, 0.5%, 3%, 15.5%, 38%, 63% and 88% cover, and in the residential garden models scores of 1–6 were converted to 5%, 17.5%, 38%, 63%, 83%, and 95% cover. Data used in the model were expressed as proportions. These conversions were to allow for more intuitive and interpretable linear results, reduce model instability caused by excessive numbers of categories, and allow for simpler aggregation of cover scores from multiple variables.

The four canopy height tiers (sub, lower, mid, high) were merged into two height strata to reduce the number of model covariates. Sub-canopy and lower canopy cover scores were averaged to form one layer named lower canopy cover (2–12 m). Mid-canopy and high canopy cover scores were averaged to form upper canopy cover (≥12 m). The remaining height tiers, herb layer and shrub layer, remained unchanged.

### 2.5. Statistical Analysis

Linear models were fitted using a Bayesian framework, with separate models fitted for rats, mice, hedgehogs, and possums. Hierarchical models were created to account for dependencies in the data caused by experimental design, i.e., *season/ city/ habitat type/ method/ line/ station* (where / indicates nesting). The effects of *season, city, habitat type* and *method* were treated as fixed effects including all two-way interactions. *Line* and *station* were added as nested random effects to account for associated variation in the observed data.

The response variable was binary, indicating detected or not detected. A logit link was used as it produces easy-to-interpret results in the form of odds and odds ratios [85]. We report three main outputs; the odds ratio (OR), credible intervals (CI, calculated as the highest density interval [HDI]), and the maximum probability of an effect (MPE). MPE is the probability that a parameter is positive or negative and corresponds to a frequentist *p* value; e.g., MPE 0.95 is equivalent to 2-tailed *p* = 0.1 [86]. The CI level was set to 90% owing to increased instability at higher ranges [87]. A covariate was deemed statistically significant if the associated CI did not include 1.

The OR represents the change between category levels for a 1-unit increase in a covariate, given all other parameters remain constant. Because most covariates in the micro-habitat data were proportions, corresponding ORs represent the change in odds between 0% cover and 100% cover. However, as it is unlikely that such large differences between stations would be observed, these ORs may be misleading with regards to observable effect size. For this reason, results for proportional variables have also been presented in their practical importance range. We defined the range of practical importance as the OR between the lower and upper quartile data ranges of each covariate. This generates an OR for differences more likely to be observed between stations when applying the model to new data; it also serves as a determination of relative effect size.

#### 2.5.1. Process of Variable Selection and Model Building

Before models were fitted, variable selection was carried out to identify the most practical and informative covariates to use—see Appendix A for details. These variables (Table 1) were then used in base models for each mammal species to enable continued variable selection towards reduced and more applicable models (see Appendix A for details): they included 11 continuous habitat covariates and six categorical variables (season, city, habitat type, method of detection, transect line, and station number). The parameters specific to a habitat or city (i.e., distance to forest, specific to residential gardens and amenity; distance to coast, specific to Wellington and Dunedin; Table 1) only affected the odds within that factor level; e.g., residential distance to forest had no effect within amenity parks. These subsets were used instead of interaction terms, as some interactions were not sensible, e.g., distance to forest within forest.

Reduced models for each species were built by retaining only variables with >85% MPE, in addition to the random effects of line and station. This process avoided excluding potentially important variables that were previously diluted by model complexity. The resulting reduced model was then fitted to the same data. The two models for each species, base and reduced, were compared using leave-one-out cross-validation (LOO CV), to determine the more robust and generalisable model, which was selected as the final model (main model) for each species.

Separate residential garden models were created to accommodate the extra variables recorded only in residential properties. To maintain comparability to the multi-habitat models, residential garden models were fitted with the same variable list as the main or final model of each species, plus the following residential garden variables: proportion of vegetation cover that was native, cover of garden beds, cover of artificial hard landscaping, level of property maintenance, and presence or absence of compost (Table 1).

The Bayesian generalised linear multilevel models were fitted using the R package *brms* (v 2.11.0; The R Journal, 2018) and compared with the *loo* package (v 2.1.0). A relatively wide prior, with a normal distribution centred on 0 and a standard deviation of 5, was used in all models. This wide prior was chosen as there were no strong beliefs about the parameter values, and it enabled the model to be influenced more by the data without strong shrinkage towards 0. A prior sensitivity analysis was carried out to assess the relative effect of prior choice on the posterior distribution. 

#### 2.5.2. Main Model Presentation 

Main models were summarised using the *bayestestR* package (v. 0.5.0) to describe effects, uncertainty, and significance within the Bayesian framework [86]. Post-hoc comparisons based on estimated marginal means were computed using the *emmeans* package (v.1.4.1). Linear model data visualisation was plotted using *sjPlot* (v. 2.7.2) and *bayesplot* (1.7.0). Conditional R^2^ and marginal R^2^ were calculated as in [88]: conditional R^2^ returns the variance explained by the model, including variance explained by random effects, while marginal R^2^ calculates the variance independent of random effects (i.e., explained by fixed effects only).

When comparing effects of city and habitat type with estimated marginal means, the results were not used to determine whether the overall detection rates differed between factor levels and interaction combinations. Instead, these comparisons asked whether, after accounting for all covariate effects in the model, there were differences between these factor levels that were unexplained by the other model parameters. For residential garden models, we focus on the additional variables specific to residential gardens. Coefficients that are shared between the main and residential garden models differ in size between these model pairs, as they were estimated from fewer data in the residential garden models, resulting in stronger or weaker estimated effects. 

Traceplots and density plots were visually assessed to check chain convergence, mixing and posterior normality. Autocorrelation plots were used to assess the extent of serial correlation within parameters. Gelman and Rubin’s potential scale reduction factor (Rhat), was used to provide a statistical estimate for convergence and variance within chains, and the effective sample size (ESS) was also noted [89]. For each of the main models, a posterior predictive check was used to assess how well the mean detection rate in each season, city, and habitat combination was predicted. 

Our final objective was to test the broader generality of our species-specific main models by applying them to two additional cities (Tauranga, 137.1 km^2^, in May 2019 and New Plymouth, 75.5 km^2^, in June 2019; Figure 1). Micro-habitat vegetation variables were not collected in these cities. Sample methods and effort were identical to those in Dunedin and Hamilton for this single sampling session, except there was no one-night deployment of a tracking card or chew-card ahead of the six-night deployment and hence cameras (where used) were also deployed for six nights not seven. The ‘two-city’ models based on these data used the same hierarchical structure as the previous models, but the only continuous predictors were distance covariates.

## 3. Results

### 3.1. Detections in Hamilton, Wellington and Dunedin

Across all three cities, all five of our predator species of interest (hedgehog, possum, mouse, black rat, brown rat) were detected during the two monitoring periods; four species (all but brown rat) were detected in all cities. In addition, cats *Felis catus*, and to a lesser extent dogs *Canis lupus familiaris*, were detected frequently, and one ferret *Mustela furo* was detected in a Hamilton forest fragment. There were also several probable mustelid detections, five in Wellington (two in residential garden habitat, two in amenity parks, and one in a forest fragment), and one in residential Dunedin. Out of 288 possible site detections per city (568 in Wellington), rats (both species combined) were detected 43 times in Dunedin, 68 times in Hamilton, and 209 times in Wellington. Respective values for other taxa were hedgehog 160, 73, 138; mouse 70, 60, 201; possum 131, 39, 1. Regarding habitat types, rats were detected 122 times in forest fragments (out of 382 possible detections), 117 times in amenity parks, and 81 times in residential gardens. Respective values for other taxa were hedgehog 106, 134, 131; mouse 107, 149, 75; possum 123, 33, 15.

Of 1147 bursts of three sequential camera images with rats identified in at least one image, 48% were black rats, 3% were brown rats, and the remaining 49% could not be identified to species. Brown rats appeared in 11% of bursts of images in Wellington residential areas, but in <4% of bursts in other cities and habitats. No brown rats were identified in Dunedin. The mean numbers of days on which cats and dogs were detected on sampling transects are provided in Appendix A. 

### 3.2. Species Detection Models (=Main Models)

After accounting for differences in micro- and landscape-scale covariates, all taxa apart from mice had significantly greater odds of being detected with cameras than with cards. Rats and mice were most likely to be detected in autumn; possums and hedgehogs in spring (Figure 2). Rats and mice had higher odds of being detected in Wellington; hedgehogs and possums in Dunedin. At a coarse scale, odds of rat detection were similar in all habitats, odds were lowest for mice in residential areas, hedgehogs were less likely to be detected in forest habitats, and possums significantly more likely to be detected in forest. 

Main models predicted mean species’ detection rates well (Figure 2), and best in the city–habitat combinations where detection numbers or sample sizes were highest. The main model for rat detections explained 22% of the variance in the data when excluding random effects (marginal [m] R^2^) and 37% when including random effects (conditional [c] R^2^); corresponding values for the residential garden rat model were 19% and 31%. The mouse and hedgehog main models explained a similar amount of variance; main hedgehog model mR^2^ = 20%, cR^2^ = 33%; main mouse model mR^2^ = 21%, cR^2^ = 33%. The residential garden hedgehog model performed better at mR^2^ = 27%, cR^2^ = 46%, while the residential garden mouse model was similar (mR^2^ = 19%, cR^2^ = 32%). The main possum model performed the best (mR^2^ = 58%, cR^2^ = 65%); there was no residential garden-specific possum model. For comparison of shared variables, the full model summaries can be found in the Appendix A.

### 3.3. Effects of Season, City, Habitat Type, and Method on Odds of Detecting Mammalian Predators

Sampling season had a significant effect on the detection of all four predator taxa (Figure 2). Rodent detections were significantly more likely in autumn than in spring, with rat detection odds 222% higher in autumn and mouse detection odds 426% higher (Table 2 and Table 4). Conversely, detection odds significantly decreased in autumn for hedgehogs (69%: Table 3) and possums (58%; Table 5). These seasonal effects, and those of city, habitat type, and their interactions with each other (below), were estimated after adjusting for all covariates (distance to forest and field variables set to 0, all others to their average) and averaging over categorical predictors that were not part of the comparison.

Rat detection odds were significantly higher in Wellington than in both Dunedin (OR = 4.78, CI = 1.67–13.7, MPE = 0.99) and Hamilton (OR = 2.84, CI = 1.04–7.97, MPE = 0.96), but not significantly different between Hamilton and Dunedin. This Wellington effect (higher than expected detections given the covariates) was present in all habitat types but was statistically significant only in forest patches (Dunedin forest: OR = 7.17, CI = 1.50–34.3, MPE = 0.98; Hamilton forest: OR= 5.42, CI = 1.24–26.8, MPE = 0.96), and relative to Dunedin amenity parks (OR = 14.2, CI = 2.25–81.4, MPE = 0.99). Overall, estimated detection odds in amenity parks were lower than in residential gardens (OR = 0.22, CI = 0.05–1.05, MPE = 0.96), but this effect was statistically significant only in Dunedin (OR = 0.05, CI = 0.00–0.57, MPE = 0.98). 

Hedgehog detection odds were higher in Dunedin than in both Hamilton (OR = 6.55, CI = 2.58–16.6, MPE = 1.00) and Wellington (OR = 8.13, CI = 3.73–19.7, MPE = 1). Hedgehog detection odds were significantly lower in forests than in amenity parks (OR = 0.35, CI = 0.14–0.88, MPE = 0.97). No interaction between city and habitat was fitted in this model.

The odds of detecting mice were significantly higher in Wellington than in Hamilton (OR = 3.62, CI = 1.77–7.53, MPE = 1.00) and Dunedin (OR = 2.36, CI = 1.16–4.70, MPE = 0.98). Between habitat types, odds were significantly higher in forest patches compared to residential gardens (OR = 4.24, CI = 1.92–9.3, MPE = 1.00) and higher in amenity parks compared to residential gardens (OR = 4.69, CI = 2.18–10.3, MPE = 1.00). City differences were most apparent in residential gardens, where expected detection odds in Wellington were significantly higher than in Hamilton (OR = 10.2, CI = 2.95–38.3, MPE = 1.00) and Dunedin (OR = 4.34, CI = 1.31–14.5, MPE = 0.98). Detection odds in Wellington amenity parks were also significantly higher than in Hamilton parks (OR = 3.21, CI = 1.08–10.21, MPE = 0.96). Habitat differences were significant in Hamilton and Dunedin, where odds of detection in forests were higher than in Hamilton residential gardens (OR = 11.9, CI = 2.91–5.07, MPE = 0.99) and detections in amenity parks were also higher than in residential gardens (Hamilton: OR = 7.09, CI = 1.74–30.6, MPE = 0.99; Dunedin: OR = 6.30, CI = 1.64–24.1, MPE = 0.99). 

Possums were virtually undetected in Wellington (one detection only). The odds of detecting possums in Dunedin were significantly higher than in Hamilton (OR = 71.52, CI = 21.2–275, MPE = 1). This effect was significant in all habitat types; forest (OR = 751, CI = 99.2–6527, MPE = 1), amenity park (OR = 8.02, CI = 2.22–37.9, MPE = 0.99), residential garden (OR = 54, CI = 5.06–668, MPE = 1.00). Detection odds were significantly higher in forest habitat compared to amenity parks (OR = 42.9, CI = 9.7–186, MPE = 1) and residential gardens (OR = 3157, CI = 206–45039, MPE = 1), and were higher in amenity parks than residential gardens (OR = 70.7, CI = 5.17–966, MPE = 1.00). This difference of habitat types was present within each city: Dunedin forest/amenity parks (OR = 412, CI = 45.7–3753, MPE = 1), forest/residential gardens (OR = 11494, CI = 752–201681, MPE = 1), amenity parks/residential gardens (OR = 27, CI = 2078–285, MPE = 0.99); Hamilton forest/amenity parks (OR = 4.35, CI = 1.06–20.6, MPE = 0.96), forest/residential gardens (OR = 824, CI = 31.5–27184, MPE = 1.00), amenity parks/residential gardens (OR = 181, CI = 6.75–6962, MPE = 1.00). 

The full pairwise post hoc comparisons of city and habitat effect are listed in the Appendix A. Detection odds were significantly lower using cards than cameras for rats, hedgehogs, and possums (Table 2, Table 3 and Table 5); there was no significant difference between the methods for mice (Table 4).

### 3.4. Effects of Habitat Covariates on Odds of Detecting Mammalian Predators in Three Main Cities

Six covariates had significant effects on detection odds for the four mammal taxa (black rats and brown rats combined). Amenity stations farther from the nearest forest patch had significantly higher odds of detecting rats (Table 2) but significantly lower for hedgehogs (Table 3) and possums (Table 5). Residential garden stations farther from the nearest forest patch had significantly reduced odds of detecting rats (see Figure 3 for linear relationships of distance variables). Higher cover in the lower canopy and increasing shrub layer cover significantly increased odds of detecting rats, whereas increasing leaf litter significantly reduced odds of detection (Table 2). Increasing herb layer cover significantly increased mouse detections (Table 4). Larger tree diameter (DBH) was unimportant for all species. 

In the residential habitat-specific models, only rat detections were significantly related to the additional residential garden parameters. Rat detection odds were significantly less in properties with a high proportion of native vegetation (Table 2; Figure 4a) and higher in properties with a compost heap. 

### 3.5. Practical Importance of Habitat Covariates

*Rat detection:* At the practical level (change in odds between the lower and upper quartile values of the covariate data), more vegetation cover in the shrub and lower canopy layers increased the odds of detecting rats by 32% and 60% respectively (Figure 4a). High leaf litter cover reduced odds by 39%. In residential gardens, an increasing proportion of native vegetation (from 5% to 63%) reduced odds of detecting rats by 69% (Figure 4a).

Mouse, hedgehog and possum detection: At the practical level, greater vegetation cover in the lower canopy layer (2–12m) increased hedgehog detection odds by 29% (Figure 4b). Greater vegetation cover in the herb layer increased mouse detection odds by 30%, whereas increased vegetation cover in the lower canopy layer (from 8% to 39%) decreased mouse detection odds by 27% and increased leaf litter cover (from 3% to 63%) reduced odds by 36% (Figure 4c). Larger tree diameter (from 10 cm to 35 cm) resulted in a 45% increase in detection odds for possums, although this effect was not significant (Figure 4d).

### 3.6. Two-City Models

In general, patterns in differences in detection odds between habitats were like those observed in main models, but models for rats showed the largest differences (Table 6). Similar to the main model, rat detection odds were significantly higher in forest patches than in residential gardens (New Plymouth: OR = 2250, CI = 7.66–4.55 × 10^5^, MPE = 0.99 Tauranga: OR = 2654, CI = 3.63–2.33 × 10^6^, MPE = 0.98). However, rat detection odds were higher in amenity parks than in residential gardens in the two-city model (New Plymouth: OR = 4.25 × 10^4^, CI = 45.6–3.67 × 10^7^, MPE = 1.00; Tauranga: OR = 7.70 × 10^5^, CI = 19.2–5.44 × 10^8^, MPE = 0.99), whereas the opposite was true in the rat main model. No significant effects of habitat or city were found for hedgehogs (Table 7). Results from the two-city mouse model (Table 8) were like those from the main model, with detection odds in forest patches and amenity parks higher than in residential gardens. These effects were significant in Tauranga (forest: OR = 3482, CI = 11.8–1.10 × 10^7^, MPE = 0.99; amenity: OR = 29538, CI = 40.3–4.20 × 10^7^, MPE = 1.00). In New Plymouth, only amenity parks had significantly higher odds than residential gardens (OR = 1805, CI = 8.2–6.26 × 10^5^, MPE = 0.99). However, amenity parks had lower odds than forest patches (OR = 0.01, CI = 0–0.69, MPE = 0.96), a result not observed in the main model. There were no significant effects between habitat types found for possums (Table 9). The full pairwise post hoc comparisons of city and habitat effect are listed in the Appendix A.

Distance to nearest field was the only significant continuous predictor in any of the two-city models, displaying a positive relationship with possum detection odds (Table 9; OR = 526, CI = 3.17–1.35 × 10^5^, MPE = 0.97) a result not observed in the main possum model. The modified species detection models, applied to the data of the additional two cities (Tauranga and New Plymouth), had mixed success. The rodent models explained the most marginal variance in the data (rat: mR^2^ = 26%, cR^2^ = 83%; mouse: mR^2^ = 35%, cR^2^ =78%); but the hedgehog and possum models had very low marginal R^2^ values (hedgehog: mR^2^ = 0.1%, cR^2^ = 63%; possum: mR^2^ = 5%, cR^2^ = 42%). 

## 4. Discussion

### 4.1. Broad-Scale Predictors of Mammal Presence

The four mammalian predator taxa of interest (rats, hedgehogs, mice, and brushtail possums) were each detected using tracking tunnels and chew cards in all three urban green space types (forest patch, amenity parks and residential gardens) and in all three cities. Cameras detected an additional three species (cats, dogs, ferrets) and allowed black rats to be distinguished from brown rats in many instances. Mustelids were detected at extremely low rates: one confirmed ferret in forest and some probable mustelid detections in residential gardens and amenity parks. Vegetation structure, proximity to forest fragments, season, city and the type of green space all influenced whether rats, mice, possums and hedgehogs were detected, but in different ways. At the broadest scale, differences in the frequency of species’ detections in different cities likely reflected latitudinal variation in climate, extent of vegetation cover across the city, resource availability, and local council policies regarding pest control. Rat and mouse detection rates were higher in Wellington than in the other two cities across the three green space habitat types sampled, whereas possums were virtually undetectable in this city. Effective possum control over several decades in Wellington (e.g., [90,91]) is no doubt responsible for the very low possum detections there and may have decreased food competition with black rats, allowing their population density to increase [92,93]. It should be noted that most research on urban rats around the world has focused on brown rats [30], whereas in New Zealand the common urban rat is the black rat and consequently our discussion focuses on this species. A tendency for black rats to be less abundant at lower, cooler latitudes [94,95] and the higher possum density could explain the lower detection rate of rats in the southernmost city, Dunedin. Low temperatures at lower latitudes have a stronger limiting effect on rat populations than on mice [95]. Possum and hedgehog detection rates were higher in Dunedin than in Wellington and Hamilton across all three types of green space. Variable patterns in the suite of predators across cities suggest that our results reflect conditions at particular locations and under particular circumstances and do not necessarily indicate what could occur at other locations.

Season contributed to variations in detections of possums, hedgehogs, black rats and mice, with higher rodent detections in autumn, likely due to seasonal population increases resulting from spring and summer breeding [27,32,56,96]. Hedgehog detections were lower in autumn, possibly coinciding with a period of reduced activity leading up to winter hibernation. The reduction in possum detectability in autumn was much less pronounced than the seasonal changes in other species.

Connectivity was also associated with predator detection in this study. Urban landscapes are typically highly heterogeneous, with varying degrees of connectivity between different types of green spaces. The degree of connectedness between green spaces is known to influence the composition of small mammal communities [13,97,98]. 

### 4.2. Habitat Associations: Rats

Black rats in New Zealand are most abundant in lowland podocarp–hardwood forest and tend to be less abundant in more open habitats [32]. Both black and brown rats can be commensal and are likely to benefit from food and shelter available in residential areas [27,32]. In this study total detections of rats were less in residential areas, possibly because they are more likely to be hunted by cats and trapped by people, however rats were more likely to be detected in residential gardens located closer to forest patches. As distance to forest patches increased more rats were detected in amenity parks. Black and brown rat populations may spill over from forest patches, where rats were apparently most abundant, into adjacent residential gardens containing resources for rats (e.g., compost). Higher rat detections in amenity green spaces situated further from forest patches is difficult to interpret and could result from landscape features we did not measure that could affect the movement of rats through the landscape and the suitability of habitat. For example, a stream, present at the edge of at least one amenity space in the study, could be associated with higher brown rat detections as they are often found in wet habitats [26,50]. Landscape features such as roads and “resource deserts” do pose barriers to rat movement across cities, although genetic analyses reveal that barriers are more permeable than previously thought (reviewed in [30]).

In terms of fine-scale habitat, rats were more likely to be detected as cover in the lower canopy and the shrub layer increased. The shrub layer, which ranged from 0.5–2 m high, may provide not only cover for rats from predators [99] but also extra feeding habitat. Black rats in particular are adept climbers, allowing them to easily access the surrounding vegetation for food [100,101,102]. Black rats in Australian forest are associated with dense understory cover and abundant vertical stems [103], and dense understory mediates negative edge effects on black rat populations in rural forest fragments in New Zealand [104]. Urban brown rats in Europe and South America display preferences for high cover of low and mid-height vegetation [105,106], and vegetation density was a positive predictor of brown rat presence in Salzburg, Austria [105]. Black rats in Australian forest preferentially chose sites with dense leaf litter [103], which was interpreted as a form of habitat complexity. However, we did not find an association between rat presence and leaf litter, and nor did a study in the Caribbean [107].

In residential gardens, rats (black and brown rats combined) were most likely to be found in gardens with compost heaps and fewer native plants. Although most of our detections were of black rats, the presence of a compost heap has been positively associated with both brown and black rat presence [108,109], serving as a food resource, but also as a home refuge. Organic waste forms the most significant food source for brown rats in urban areas [108]. Well-kept areas, such as gardens, have been associated with reduced abundance of brown rats [58]. While properties in England with unkempt gardens had a significantly higher prevalence of brown rats, property maintenance did not significantly influence rat detections in our study. An increasing proportion of native vegetative cover had a strong negative effect on rat detections in residential gardens. Since black rats are often present in high numbers in the native bush of New Zealand [92,100,110], it seems unlikely that this is due to an aversion to native plant species but could be a consequence of the behaviour of the residential property owners. Property owners who place high value on native diversity may also be inclined to engage in other conservation values, such as pest control [111], although there is not always a positive relationship between native plant diversity in residential gardens and strong ecological values of householders [75].

### 4.3. Habitat Associations: Hedgehogs

In New Zealand, hedgehogs do not always show strong preferences between urban habitat types [31]. We did not find a preference for residential gardens, which typically have a higher density of structures that have been shown in Europe to be associated with hedgehogs [13]. Hedgehogs were less likely to be detected in forests than in amenity parks, which comprise mostly lawn. Lawn can support high earthworm biomass, which in France was a good predictor of hedgehog abundance [112]. It is difficult to explain why fewer hedgehogs were detected in amenity parks further from forest fragments, which did not appear in the validation data from Tauranga and New Plymouth. Habitat connectivity influences the movement of urban hedgehogs in Europe but is considered less important than habitat quality [13]. Hedgehogs will move across all green spaces and apparently impervious areas with little vehicular traffic, although main streets act as major movement barriers [113]. The presence of roads with high traffic volume could explain patterns of distribution in this study, and these influences on connectivity should be measured and included in future models.

We found no evidence of an association with lawns and garden bed cover, lower canopy cover and compost. Non-discriminatory behaviour by urban hedgehogs in relation to compost heaps has been observed previously [114]. Compost presence in this study was a combined grouping of compost heaps and compost bins, the latter being less accessible to hedgehogs than to rodents.

### 4.4. Habitat Associations: Mice

Mice occur in a wide range of habitats in New Zealand, including different types of forest, but also edge habitats and rank grass [56]. Mice were detected most often in amenity parks, and least often in residential gardens, where they are hunted by cats and trapped by people. In residential gardens there was no association between mice and compost heaps, the level of property maintenance, or the proportion of native vegetation. While mice might also benefit from resources such as compost heaps, their absence could be due to interspecific competition between the three rodent species; rats are bigger and can be aggressive competitors and potentially direct predators on mice [115,116,117]. Distance to forest patches or open habitat did not explain variation in mouse detections.

Mouse detections were positively associated with increasing herb layer. House mice tend to reach higher population densities in areas with dense ground cover in New Zealand [36] and in natural reserves and parkland in Buenos Aires [110]. Vegetation cover at a low height level likely gives individuals a sense of security, as they can more easily hide from potential predators [118]. In New Zealand beech forests, mice were linked with increased leaf litter [119], which supports a higher density of arthropods that frequently make up a large component of mouse diet. However, in our study mice were not detected more often at sites with higher leaf litter cover. We recorded habitat characteristics in late autumn by which time deciduous trees had lost much of their foliage, forming areas of dense leaf litter; these site values were applied to data collected in spring as well, which could have obscured potential associations.

### 4.5. Habitat Associations: Possums

Urban possums are known to occupy forest fragments [31,120] where there are ample food resources (foliage) and den sites [121,122], compared with other habitats. We detected possums most often in the forest fragments, and least often in residential gardens. In Dunedin, possums were detected at all stations in forest fragments; at one of these (Jubilee Park), possum densities have been estimated at approximately 3 ha^−1^, thirty times higher than in nearby residential areas, and similar to densities estimated in native southern beech forest [123]. Although urban possums prefer forest fragments, they regularly make forays into residential properties and can even have home ranges across residential gardens that are completely independent of forest fragments, providing the gardens have mature, structurally complex vegetation [120,124]. All our residential possum detections occurred in properties with such gardens; however, the presence of dogs may discourage possums from occupying gardens. Connectivity played a role in influencing distributions in that fewer possums were likely to be detected in amenity parks as distance to forest increased. 

Although tree size (DBH) has been identified as an important component of habitat selection by brushtail possums in urban parkland in Australia [125], possums can permanently inhabit non-forested areas, such as scrub and grassland [126,127], and this likely resulted in a non-significant association with DBH in our study. 

### 4.6. Application to Two Further Cities

The two-city models, which did not include fine-scale habitat variables, explained predator presence poorly across all species. Only one of the landscape-scale variables was significant (distance to nearest field), and only for possums. The very low marginal R^2^ values for hedgehogs and possums can be attributed to sample size, as there were fewer detections of these species (20 and 12 respectively) than for rats and mice (70 and 98 respectively). Conditional R^2^ values were higher than in the main models, indicating more variance attributed to the random effects of site and station. Credible intervals were generally wide in the two-city models owing to the relatively small sample (two cities sampled in one season, compared with three cities sampled in two seasons in the main models). These results suggest that large sample sizes are needed for this type of analysis and to accommodate heterogeneity between cities, and that unmeasured variables may have been relatively influential in these two cities. 

### 4.7. Model Performance and Study Limitations

The possum model explained more than 50% of the variation in the data. Random variation between sites had little effect on this model’s predictive success, likely owing to the strong association of possums with forest patches, although the near saturation of possum detections in Dunedin forest patches is likely to have over-ridden any effects of other predictors. The main models of the remaining three taxa explained between 33% and 37% of variation. They were also relatively reliant on the variance explained by random effects, as removing these reduced the explanatory power of these models by >11%, indicating that some consistent trends were not explained by the model covariates. 

The most widely used detection methods in this study (chew cards and tracking tunnels) cannot distinguish between species of rat. As a consequence, the rat model was generalised over black and brown rats, which differ in behaviour and habitat selection [110,115,128]. For example, the preference of brown rats for wet habitats [27,108] is not shared by black rats. The absence of the variable “distance to water bodies” from the final reduced rat model may reflect this difference, and that black rats dominated our rat detections.

In any future assessment of habitat preference, the presence of fruiting tree species could be worth recording. Fleshy fruits can make up a large proportion of the diet of rats and possums in natural and rural habitats [92,119]. In rural habitats possums will significantly extend their home ranges to exploit fruit trees [129] and in urban areas of Australia possums access residential properties for fruit, vegetables, and compost [124]. Additional landscape features, such as road density or traffic volume, could also improve the explanatory power of the models. Mustelids are clearly not abundant in urban areas, perhaps because rabbits (main prey of ferrets) are sparse and because stoats and weasels are preyed on by the abundant cats [130,131,132,133]. The trail cameras and Erayz lures used in this study are promising techniques for detecting these species in New Zealand [134], as confirmed using alternative detection methods [135].

### 4.8. Towards More Efficient Urban Predator Control

Our results suggest that control efforts should be targeting the full suite of common mammalian predators to reduce potentially adverse outcomes resulting from competitive release, such as an increase in rat abundance after possum control [93]. Possums are likely to be the most easily controlled species, given their strong association with forest patches, low reproductive rate [136], and the success of previous urban possum control operations; for example, on Wellington’s Miramar Peninsula [137]. Our results suggest that possum control may be beneficial in parks and well-vegetated residential gardens as well. 

Hedgehogs would require widespread control, as they displayed no strong associations with any of the habitat variables. Because hedgehogs were the species most likely to be detected in gardens, residential property owners could be effective at reducing urban hedgehog populations if they participated in control efforts, which could be focused during spring when hedgehogs emerge from hibernation. Rodent detections were lowest in residential gardens, suggesting control should focus mostly on forest and amenity park habitat, with traps placed in areas with high quantities of low vegetation cover. Residential property owners should also be encouraged to use rodent-proof composting methods to limit rodents’ access to valuable food resources. 

## 5. Conclusions

This study confirms that while habitat associations vary between species, the five invasive predatory mammals we studied are common in a range of urban landscape types, including residential gardens, indicating the need for widespread and coordinated control operations to support initiatives aimed at restoring native biodiversity. The presence of all five pest taxa in residential gardens poses a further challenge requiring the engagement of residents in order that these habitats do not support populations acting as sources of reinvasion into other green spaces.

## Figures and Tables

**Figure 1 biology-11-01527-f001:**
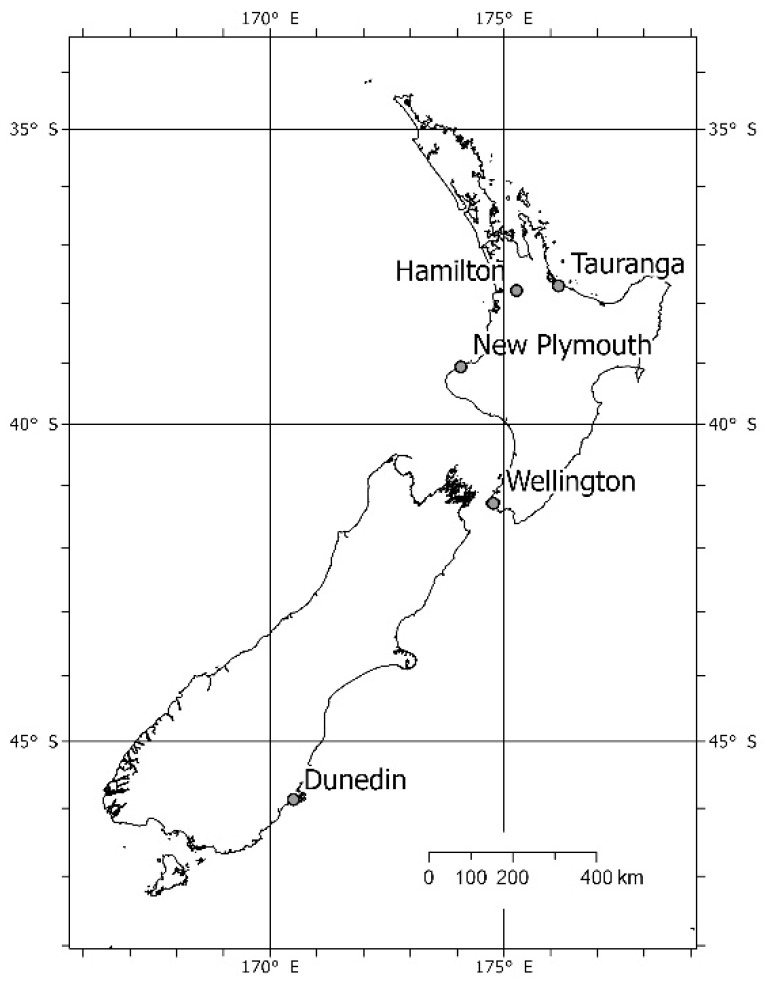
Map of New Zealand showing cities surveyed.

**Figure 2 biology-11-01527-f002:**
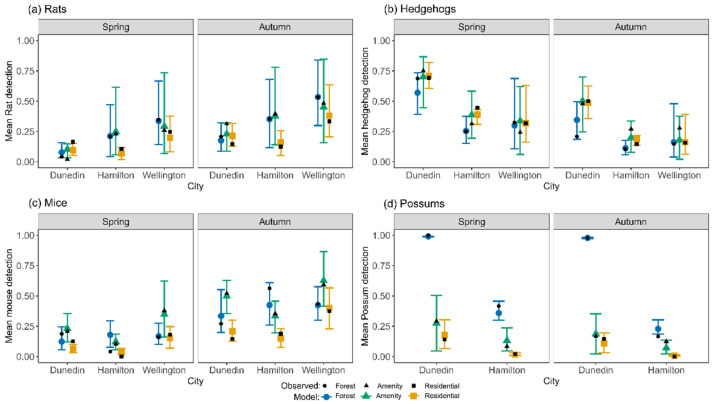
Detection probability of mammalian predators across city and habitat type, split by season, for (**a**) rats, (**b**) hedgehogs, (**c**) mice, and (**d**) possums. Black symbols represent observed mean detection rate; coloured symbols are mean estimated probability of detection based on predictions of the main model. Bars represent 90% credible intervals. Fitted probabilities for each sampling location (from the posterior distribution) were averaged within each season, city, and habitat type and across devices (i.e., 12 possible detections per line from 10 stations and two cameras). Possums were detected in Wellington only once in spring (mean detection = 0.001) and were not modelled.

**Figure 3 biology-11-01527-f003:**
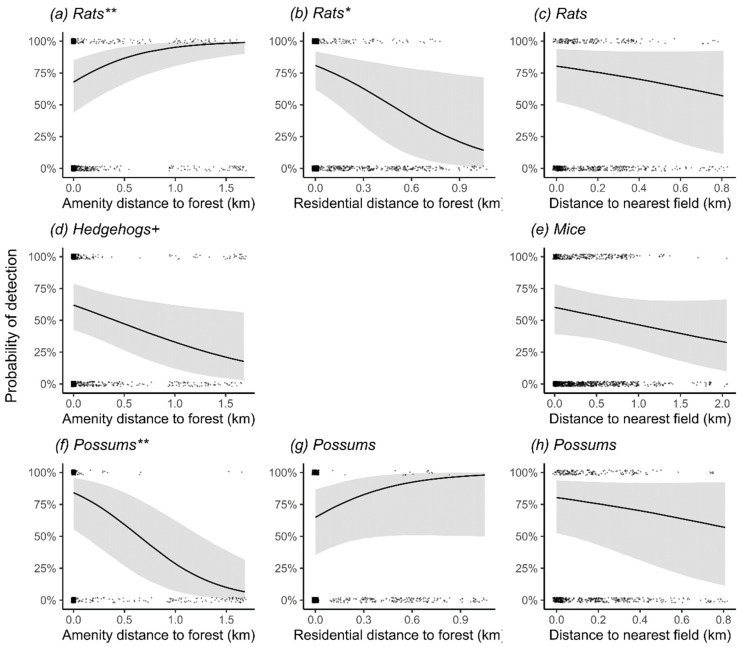
Model covariate effect plots showing change in predicted detection probability (90% CI shaded) of rats, hedgehogs, mice and possums in relation to distance covariates (km). Black dots are the raw data values of the response variable, indicating detection (top of each graph, representing 1), and non-detection (bottom, representing 0). Data for rodents are conditioned on Season = Autumn, City = Wellington, Habitat type = Forest, and Method = Camera. For hedgehogs and possums, City = Dunedin. All other continuous covariates are set to mean values. As there were no interaction terms between covariates in these models, the plotted relationships are constant across all factor combinations, with only the intercept position changing. +, * and ** indicate statistical significance as defined in Table 2.

**Figure 4 biology-11-01527-f004:**
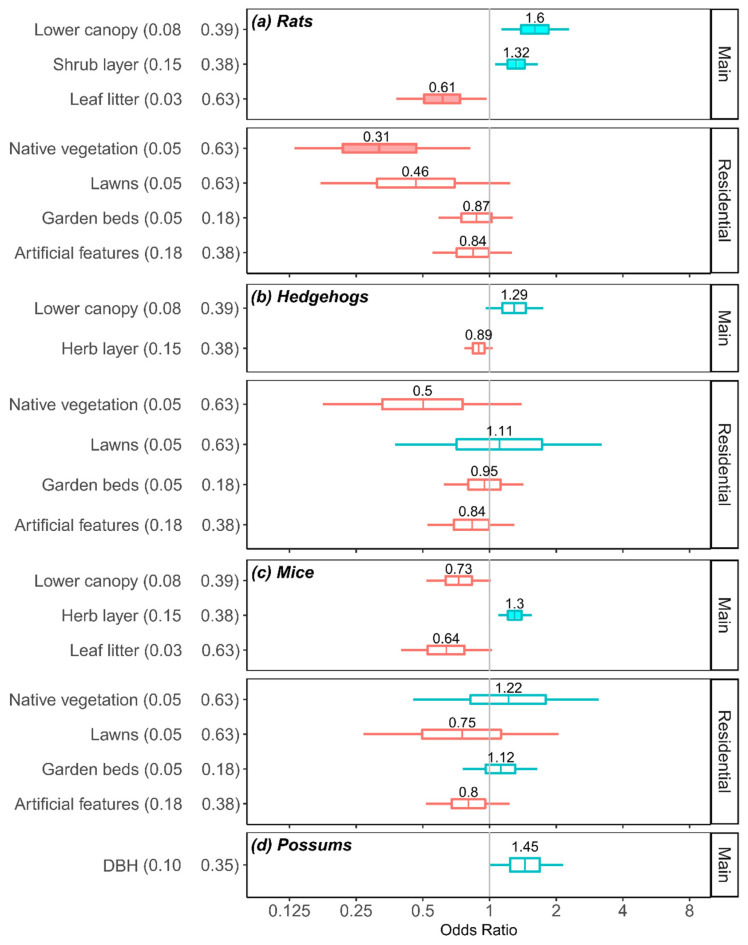
Practical significance of continuous covariates in the main models and residential models of four mammalian taxa: (**a**) rats, (**b**) hedgehogs, (**c**), mice, (**d**) possums (no residential model fitted). Boxplots show the posterior distribution of odds ratios (OR) for a predicted change in detection odds. The OR are estimated from the difference in predicted detection odds at the 25% vs. 75% quartiles of each covariate; these quartile values are given after each parameter name. Median OR displayed above each box, interquartile range is 25–75% of distribution, and tails cover 90% of the distribution (5–95%). The grey vertical band lies at OR 1 (no effect), blue distributions are net positive effects, red distributions signify negative relationships, and shaded boxes represent significant effects (i.e., 90%CI of OR does not include 1).

**Table 1 biology-11-01527-t001:** List of parameters and their units in the base models for each mammal species. Proportion data range between 0 and 1.

Parameter Name	Description
*Season*	Sampling season: spring or autumn (2 levels, categorical)
*City*	City: Hamilton, Wellington, Dunedin (3 levels, categorical)
*Habitat type*	Forest fragment, Amenity park, Residential garden (3 levels, categorical)
*Method*	Method of detection: Cards (tracking tunnels, chew cards) or camera (2 levels, categorical)
*Distance to nearest field*	Distance from sampling station to nearest grassy field > 1 ha (km)
*Distance to coast (Dunedin)*	Distance to the coast in Dunedin. (Rat and mouse models only) (km)
*Distance to coast (Wellington)*	Distance to the coast in Wellington. (Rat and mouse models only) (km)
*Distance to freshwater*	Distance to nearest freshwater body. (Rat and mouse models only) (km)
*Residential distance to forest*	Distance of residential sites to nearest forest fragment > 1 ha (km)
*Amenity distance to forest*	Distance from amenity park sites to nearest forest fragment > 1 ha (km)
*DBH*	Diameter at breast height of the largest tree in sampling plot (m)
*Leaf litter cover*	Proportion of ground area covered by leaf litter.
*Herb layer cover*	Total vegetation cover within the herb layer (0–0.3 m high)
*Shrub layer cover*	Total vegetation cover within the shrub layer (0.3–2 m high)
*Lower canopy cover*	Average total vegetation cover in sub- and lower canopy layers (2–12 m high)
*Line*	Transect line identifier (random effect, 48 levels, categorical)
*Station*	Station within each line (random effect, 10 levels, categorical)
**Residential only** **parameters**	
*Compost*	Presence/absence compost heap/bin (2 levels, categorical)
*Level of property maintain*	Level of property maintenance: low/medium/high (3 levels, categorical)
*Native vegetation*	Proportion native vegetation cover
*Lawns*	Proportion cover by lawns
*Garden bed cover*	Proportion cover of regularly turned soil beds i.e., flower & vegetable beds
*Artificial features*	Proportion cover of artificial hard landscaping features i.e., buildings, decking, paving, fences

**Table 2 biology-11-01527-t002:** Rats: summary statistics for coefficients of the parameters in the main rat model and the additional residential model parameters.

Parameter	OR	CI	MPE	Significance
Intercept	0.46	0.13–1.54	0.857	
**Season = Autumn**	**3.22**	**2.36–4.27**	**1**	******
**City = Hamilton**	**0.18**	**0.04–0.80**	**0.966**	**+**
**City = Dunedin**	**0.14**	**0.03–0.67**	**0.982**	*****
Habitat = Amenity	0.46	0.13–1.67	0.845	
Habitat = Residential	0.63	0.16–2.29	0.714	
**Method= Cards**	**0.34**	**0.23–0.50**	**1**	******
Hamilton: Amenity	1.55	0.20–13.7	0.632	
Dunedin: Amenity	0.49	0.05–4.80	0.699	
Hamilton: Residential	4.46	0.47–50.2	0.853	
Dunedin: Residential	6.61	0.75–55.4	0.930	
Distance to nearest field	2.08	0.76–5.89	0.881	
**Residential distance to forest**	**0.05**	**0–0.58**	**0.982**	*****
**Amenity distance to forest**	**9.29**	**2.63–34.43**	**0.998**	******
**Shrub layer cover**	**3.42**	**1.27–9.03**	**0.981**	*****
**Lower canopy cover**	**4.51**	**1.43–13.54**	**0.988**	*****
**Leaf litter**	**0.44**	**0.20–0.95**	**0.960**	**+**
Residential Model				
Level of maintenance = medium	0.43	0.14–1.16	0.918	
Level of maintenance = high	0.39	0.12–1.31	0.909	
**Compost = Yes**	**2.77**	**1.19–6.25**	**0.986**	*****
**Proportion of native vegetation**	**0.13**	**0.02–0.71**	**0.977**	*****
Mown lawn cover	0.26	0.04–1.41	0.902	
Garden bed cover	0.34	0.01–6.58	0.716	
Artificial characteristics	0.43	0.05–3.03	0.760	

Odds ratio (OR) is calculated at the median posterior density. Credible intervals (CI) are highest posterior density intervals (HDI) at the 90% level (5%–95%; used because of instability at higher levels). MPE is maximum probability of effect. Statistical significance (when the 90% CI does not include 1) is denoted by + and bolded. MPE was always >0.95 (equivalent to frequentist 2-tailed *p* < 0.1) when the 90% CI did not include 1. MPE ≥0.975 and ≥0.995 (equivalent to frequentist 2-tailed *p* < 0.05 and *p* < 0.01) are shown as * and **, respectively. Effects of categorical variables are relative to a reference category, i.e., Season = Spring, City = Wellington, Habitat type = Forest fragment, Method = Camera, Level of maintenance = Low, and Compost = No. Interactions indicated by “:”.

**Table 3 biology-11-01527-t003:** Hedgehogs: summary statistics for coefficients of the parameters in the main hedgehog model and additional residential model parameters.

Parameter	OR	CI	MPE	Significance
**Intercept**	**0.33**	**0.13–0.78**	**0.982**	*****
**Season = Autumn**	**0.31**	**0.23–0.41**	**1**	******
City = Hamilton	1.23	0.55–2.89	0.666	
**City= Dunedin**	**8.24**	**3.61–19.7**	**1**	******
**Habitat = Amenity**	**2.86**	**1.11–7.10**	**0.969**	**+**
Habitat = Residential	1.89	0.84–4.47	0.902	
**Method = Cards**	**0.56**	**0.40–0.81**	**1**	******
**Amenity distance to forest**	**0.30**	**0.10–0.87**	**0.973**	**+**
Lower canopy layer	2.31	0.91–6.31	0.922	
Herb layer	0.61	0.30–1.17	0.898	
Residential Model				
Level of maintenance = medium	1.14	0.32–4.23	0.555	
Level of maintenance = high	0.98	0.24–3.98	0.512	
Compost = Yes	0.67	0.26–1.63	0.770	
Proportion of native vegetation	0.29	0.05–1.70	0.874	
Lawn cover	1.20	0.17–7.16	0.562	
Garden bed cover	0.64	0.03–18.6	0.583	
Artificial characteristics	0.40	0.04–3.92	0.754	

OR, CI, MPE and significance are defined as in Table 2. Effects of categorical variables are relative to a reference category, i.e., Season = Spring, City = Wellington, Habitat type = Forest fragment, Method = Camera, Level of maintenance = Low, and Compost = No.

**Table 4 biology-11-01527-t004:** Mice: summary statistics for coefficients of the parameters in the main mouse model, and additional residential model parameters.

Parameter	OR	CI	MPE	Significance
Intercept	0.35	0.12–1.04	0.947	
**Season = Autumn**	**5.26**	**3.8–7.01**	**1**	******
City = Hamilton	0.70	0.23–2.14	0.712	
City = Dunedin	0.52	0.17–1.59	0.838	
Habitat = Amenity	1.35	0.52–3.49	0.702	
Habitat = Residential	0.59	0.23–1.52	0.824	
Method = Cards	0.83	0.55–1.22	0.784	
Hamilton: Amenity	0.45	0.10–2.00	0.812	
Dunedin: Amenity	1.21	0.27–5.50	0.588	
**Hamilton: Residential**	**0.14**	**0.03–0.73**	**0.978**	*****
Dunedin: Residential	0.44	0.09–2.07	0.814	
Distance to nearest field	0.58	0.25–1.25	0.877	
Lower canopy cover	0.36	0.12–1.03	0.943	
Leaf litter cover	0.47	0.22–1.07	0.941	
**Herb layer cover**	**3.18**	**1.47–6.89**	**0.994**	*****
Residential Model				
Level of maintenance: medium	0.85	0.29–2.65	0.598	
Level of maintenance: high	0.45	0.12–1.60	0.844	
Compost: Yes	0.97	0.43–2.17	0.52	
Proportion of native vegetation	1.40	0.26–7.24	0.63	
Lawn cover	0.61	0.11–3.68	0.68	
Garden bed cover	2.54	0.11–54.1	0.69	
Artificial characteristics	0.34	0.05–3.13	0.80	

OR, CI, MPE and Significance are defined as in Table 2. Effects of categorical variables are relative to a reference category, i.e., Season: Spring, City = Wellington, Habitat type = Forest fragment, Method = Camera, Level of maintenance = Low, and Compost = No.

**Table 5 biology-11-01527-t005:** Possums: summary statistics for coefficients of the parameters in the main possum model.

Parameter	OR	CI	MPE	Significance
**Intercept**	**526**	**82.3–3577**	**1**	******
**Season = Autumn**	**0.41**	**0.24–0.70**	**0.998**	******
**City = Hamilton**	**0**	**0–0.01**	**0.994**	*****
**Habitat = Amenity**	**0.01**	**0–0.04**	**1**	******
**Habitat = Residential**	**0**	**0–0.00**	**1**	******
**Hamilton: Amenity**	**43.1**	**5.11–373**	**0.992**	*****
Hamilton: Residential **Method = Cards**	4.69 **0.21**	0.23–83.2 **0.10–0.43**	0.805 **1**	******
Distance to nearest field	0.25	0.03–2.34	0.854	
**Amenity distance to forest**	**0.07**	**0.02–0.27**	**1**	******
Residential distance to forest	22.9	0.87–784	0.933	
DBH	4.36	0.95–19.9	0.949	

Odds ratio (OR) is calculated at the median posterior density. OR, CI, MPE and Significance are defined as in Table 2. Effects of categorical variables are relative to a reference category, i.e., Season = Spring, City = Dunedin Habitat type = Forest fragment, and Method = Camera.

**Table 6 biology-11-01527-t006:** Rats: summary statistics for coefficients of the parameters in the two-city rat model fitted to data from New Plymouth and Tauranga.

Parameter	OR	CI	MPE	Significance
Intercept	0.47	0–38.3	0.62	
City = Tauranga	1.34	0.02–96.8	0.543	
Habitat = Amenity	19.17	0.15–2707	0.855	
**Habitat = Residential**	**0**	**0–0.13**	**0.99**	*****
Tauranga: Amenity	1.55	0.01–564	0.552	
Tauranga: Residential **Method = Cards**	0.81 **0.04**	0–284 **0–0.24**	0.524 **1**	******
Distance to nearest field	0.02	0–10.3	0.856	
Residential distance to forest	0.65	0–1001	0.54	
Amenity distance to forest	0.8	0–505	0.524	

OR, CI, MPE and significance are defined as in Table 2. Effects of categorical variables are relative to a reference category, i.e., City =New Plymouth, Habitat type = Forest fragment, and Method = Camera.

**Table 7 biology-11-01527-t007:** Hedgehogs: summary statistics for coefficients of the parameters in the two-city hedgehog model fitted to data from New Plymouth and Tauranga.

Parameter	OR	CI	MPE	Significance
**Intercept**	**0**	**0–0.01**	**1**	******
City = Tauranga	11.4	0.16–777	0.858	
Habitat = Amenity	7.41	0.04–1361	0.738	
Habitat = Residential	85.6	0.94–9633	0.949	
**Method = Cards**	**0.08**	**0.01–0.58**	**0.992**	*****
Distance to nearest field	2.57	0.01–930	0.6	

OR, CI, MPE and significance are defined as in Table 2. Effects of categorical variables are relative to a reference category, i.e., City = New Plymouth, Habitat type = Forest fragment, and Method = Camera.

**Table 8 biology-11-01527-t008:** Mice: summary statistics for coefficients of the parameters in the two-city mouse model fitted to data from New Plymouth and Tauranga.

Parameter	OR	CI	MPE	Significance
**Intercept**	**0.02**	**0–0.59**	**0.976**	*****
City = Tauranga	14.1	0.2–695	0.848	
**Habitat = Amenity**	**98.7**	**1.45–7806**	**0.962**	**+**
Habitat = Residential	0.05	0–4.33	0.876	
Tauranga: Amenity	0.09	0–28.4	0.768	
Tauranga: Residential Method = Cards	0.01 2.5	0–1.17 0.62–9.72	0.943 0.87	
Distance to nearest field	1.7	0–664	0.558	

OR, CI, MPE and significance are defined as in Table 2. Effects of categorical variables are relative to a reference category, i.e., City = New Plymouth, Habitat type = Forest fragment, and Method = Camera.

**Table 9 biology-11-01527-t009:** Possums: summary statistics for coefficients of the parameters in the two-city possum model fitted to data from New Plymouth and Tauranga.

Parameter	OR	CI	MPE	Significance
**Intercept**	**0**	**0–0.11**	**0.999**	******
City = Tauranga	1.88	0.03–142.32	0.605	
Habitat = Amenity	8.32	0.08–841.47	0.781	
Habitat = Residential	0.01	0–3.39	0.911	
Tauranga: Amenity	0.08	0–21.81	0.778	
Tauranga: Residential **Method = Cards**	0.13 **0.11**	0–176 **0.02–0.52**	0.685 **1**	******
**Distance to nearest field**	**526.39**	**3.17–134,313.31**	**0.972**	**+**
Residential distance to forest	0.16	0–393.44	0.654	
Amenity distance to forest	0.84	0–470.29	0.519	

OR, CI, MPE and significance are defined as in Table 2. Effects of categorical variables are relative to a reference category, i.e., City = New Plymouth, Habitat type = Forest fragment, and Method = Camera.

## Data Availability

The data presented in this study are openly available in DataStore Welcome—Manaaki Whenua—Landcare Research DataStore at [https://doi.org/10.7931/xxm8-1e75].

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
