# Peer review of "Invasive Urban Mammalian Predators: Distribution and Multi-Scale Habitat Selection"

_biology, 2022, doi:10.3390/biology11101527_

Round 1

Reviewer 1 Report

The manuscript: ‘Invasive urban mammalian predators: distribution and multiscale habitat selection’ offers an important contribution to the field of urban conservation. Understanding how introduced predators utilise green spaces, based on the habitat components is an important step towards implementing appropriate management to conserve native biodiversity. This manuscript does well to explore this and consider the results against those found globally. Below I have provided some suggested edits, mainly, the Introduction and Discussion sections need to be restructured and refined. I also have some concern as to why the habitat components were recorded as categories and then transformed for analyses when they could have been better assessed if in their raw direct count form (instead of categories that require further adjustments).

Summary

Line 16 and 18: be consistent with either ‘green space’ or ‘greenspace’, the former is more grammatically correct.

Abstract

Line 35: there is a double space before the sentence starting with “Distance”.

Line 39: “a less” used here does not read well, perhaps “lower” or “reduced” would be better in its place.

Introduction

I think that the introduction is well written but can be hard to follow and would benefit from some restructuring and reworking to be more specific, and to flow better. The additional information that I suggest including below will help to make it more clear in detailing: the value of urban green spaces to natives, the need to manage them owing to introduced predator presence, how this is particularly apparent in NZ, how understanding microhabitat use of introduced predatory mammals could help with management applications of green spaces with justification for your study, and the study details including research questions and brief methods and with some relevance to conservation significance and how the results may be used.

Line 47: there is a misplaced open parenthesis “(“at the start of this line.

Line 48: needs a comma after “green infrastructure”.

Line 46-49: this is a long and fragmented sentence that would read better if it were two and did not start with “while”.

Line 52: would read better as: … urban green spaces provide,…

Line 54: “(“ should be “[“.

Paragraph 2: I think it would be good here to state that NZ does not host any native mammalian predators, and to list the specific introduced predators here and how long they have been present and give some context to the extent of damage they can cause on native NZ species, before going on to discuss their utilisation of urban green spaces and how they sustain their populations. This new information is important to give context to the study, but only needs to be concise. Perhaps the information from lines 106-117 could be used around here, with the inclusion of the above mentioned information, to flow better into an argument for your research.

Line 82: it is worth noting here, or somewhere in the introduction, that urban green spaces that have a level of vegetation complexity, be it native or not (there is support for both published), and connectivity, can support a diversity of native fauna that is often abundant despite the small habitat size and additional stressors of human disturbances and introduced predators. I know that there is a brief broad mention of this in the first paragraph and more in the last paragraph, but I think it could be more specific and should all be placed at the start of the introduction. I think it is important to detail more how urban green spaces can be used by both native and introduced species, and therefore play a valuable role in sustaining biodiversity. This also requires a level of consideration in the pest management of urban areas, to select methods that will not affect natives, and that the people in these areas will accept. This may be more apparent in countries other than NZ but as this is a global journal this needs to be acknowledged somewhere in the introduction too. It does not need to take up too much word count but could perhaps be achieved through changing and slightly extending what is already there.

Line 84: as some of the species listed can be native in some countries, I think it would be good to continue to state “introduced mammals” each time, despite having stated it at the start. The suggested additional information for Paragraph 2 would also help to make this clearer.

Line 92: is a bit general and it would be good if it could be re-written, or had more included, to better support the justification of this study, particularly after the results of similar are given.

Line 96: the study did not aim, you did, please reword to either: “In this study we aimed to” or “The aim of this study was to, therefore,”.

Line 101: I think if using common names then “black rat” is more globally used than “ship rats”, and if changing to this then “brown rat” is as widely used as “Norway rat” and provides an easy distinction between the two Rattus species.

Line 102: as mustelids have not been previously mentioned, it is a bit confusing to mention them here. The suggestion made regarding Paragraph 2 should help clarify this. Further, this point needs some clarification. What exactly are “reasonable numbers”, please be more specific. Are you trying to state that you selected the “non-companion” introduced predators to study as they can be more readily managed via target trapping, unlike the introduced mustelids and domestic cats that are also present in the areas, though in smaller numbers, but are harder to manage owing to their relationships with people (as pets) and their aversions to traps? However, if this is what you are trying to state it is a bit misleading as mustelids and cats are mentioned in the methods and results.

Line 104: perhaps “prevalent” is a better word than “important” here, but as stated above I think that these two sentences could do with a rework to make your point and justification for the species monitored clearer.

The third last paragraph (96-105): as the aim of the study is first introduced here it would be good to follow on here with the research questions and how they will be briefly investigated, any predictions, then the conservation relevance that the results may have (briefly). This should be the concluding paragraph of the introduction. This will require removing the objectives listed in the below two paragraphs from the surrounding information, that as mentioned above and below would be much better placed earlier on in the Introduction, and placing all of the aims/objectives together, with some more detail to better clarify just what you are testing and how.

Line 118: This sentence would work better if flipped and made specific to the study, so that it read: We aimed to identify common features determining urban predator distributions in three cities (……) and three types of urban green space (……..). It would also be better placed with the preceding paragraph that details the aims of the study.

The last two paragraphs (106-135): as mentioned above the details given in these paragraphs would be better placed earlier on in the introduction, before the details of your study, this will help build a better flow for the need of this study. As the preceding paragraph (third from last) details the aims of the study this would be better to extend (see above) and close the introduction.

Materials and Methods

Line 140: what exactly is primary and secondary predominantly native forest?

Line 142: I think it would be good given the global audience to include here that amenity parks were “highly modified” and to also include at the end of the sentence if the vegetation patches were remnant native or introduced or both.

Line 143: as in line 141 you state that “residential gardens” are a green space type that was sampled I think that it is best to continue with this labelling, and to replace “Household properties” at line 143 with “residential gardens”.

Line 159: were the tracking tunnels baited with a food lure or not? Okay, I can see this is included at lines 181-184 and would be best to be moved up to here.

Line 160: were the cameras infrared allowing for less disruptive night vision? If so, please insert “infrared” between motion-activated and cameras.

Statistical analyses: this section would benefit from an edit, there is much detail given on how to interpret the results, which could be really refined, and the detail that is given on the parameters of the models and how they were selected then ran is spread throughout and would be better placed together. The proceeding sections on the models provide some good information but again this is all very long and should be reduced into one single section on statistical analyses. As these sections are now, they appear disjunct with cut and paste text book style information in parts that could be presented more succinctly and relative to the study, which would display a better understanding of the processes used.

Lines 218-220: I am a bit concerned that the habitat surveys were taken based on categories and then the data was transformed for analyses. Would it not have been more accurate to just leave these as the direct percent cover for each when measuring, instead of grouping into categories that are not even and then transforming them back to a mid-percent value that is assessed as a proportion, thus giving two transformations (category, and mid-percent proportion) on the actual value which may give different results to what is naturally occurring, especially at the finer scale. I think that analyses would be better ran on the raw data, then based on the findings could be related to categories if they were evident.

Line 228: this could be reworded to be a bit clearer. For example, to something like: To test for the effects of fine- and landscape-scale green space habitat components on the presence of introduced predators in NZ we fit separate linear models for rats (grouping brown and black rats together), mice, hedgehogs, and possums using the program <insert name and reference here> and the package <(or setting) insert name and reference here>. Linear hierarchical models, using a Baysian framework, were created to……

Line 264: I think the word “rationale” could be inserted here before variable selection, to reduce confusion as to whether it was a step wise statistical assessment of the relevance of each variable that was used in selection processes.

Results

This section was well presented and thorough, if anything despite the number of figures and tables given, I think that a summary visual representation of the results for each species relative to the finer-scale habitat parameters could be given together in one table or figure for easier quick reference than reading through the text, which is still adequate. However, I do wonder if the results would be different if modelled off the raw (uncategorised and unadjusted) data, as suggested above.

Discussion

The Discussion section does well to support the results but some language changes could be made to appear less bias. The Discussion section is overall very long and could do with some restructuring and refining, particularly the first section. It would flow better if each paragraph focused on an animal relative to the habitat components, instead of jumping back and forth between the two. Sections do appear to be grouped somewhat by habitat components or animal but are often confusing in their delivery, and repetitive of each other for an animal, which could be reduced if presented per animal per paragraph – as this is how it was modelled. These changes could also help in reducing the word count.

Line 561: there is a double space before the sentence starting with “Hedgehog detections”.

Line 565: from my experience brushtail possums in Australia are often present in higher densities in green space habitats that are closer to urban build-up than in the larger surrounding forest fragments, so I do not think that it is right to state “unsurprisingly” as it is really location and habitat dependent. There certainly are amble food and shelter resources in urban yards and the surrounds.

Line 586-587: I think this needs to be more speculative than certain unless you can provide evidence to support it. For example, you could instead state: This may be relative to the likelihood of mice and possums being predated upon by domestic pets, and/or traps by people (and give references).

Line 593-596: it is unclear here in which animal you are referring to, or if none of them found a preference for residential gardens. Further, the run-on of this sentence is confusing. Why is it then related to amenity parks? This should be two separate sentences. As it is written now it is not very clear why you are linking these two points together.

Line 646: there is a double space before the sentence starting with “The shrub layer”.

Section 4.3: I think it should also be noted that differences could be relative to other factors that were more influential for these areas that were not measured in this study. This highlights the importance for the need to clarify throughout the discussion that the results were relative to that given location under the observed conditions and may only be suggestive of what could occur at other locations.

Author Response

Reviewer 1

Comments and Suggestions for Authors

The manuscript: ‘Invasive urban mammalian predators: distribution and multiscale habitat selection’ offers an important contribution to the field of urban conservation. Understanding how introduced predators utilise green spaces, based on the habitat components is an important step towards implementing appropriate management to conserve native biodiversity. This manuscript does well to explore this and consider the results against those found globally. Below I have provided some suggested edits, mainly, the Introduction and Discussion sections need to be restructured and refined. I also have some concern as to why the habitat components were recorded as categories and then transformed for analyses when they could have been better assessed if in their raw direct count form (instead of categories that require further adjustments).

We wish to thank the reviewer for the constructive criticism provided which we feel has improved the manuscript.

 Summary

Line 16 and 18: be consistent with either ‘green space’ or ‘greenspace’, the former is more grammatically correct.

This has been corrected.

 Abstract

Line 35: there is a double space before the sentence starting with “Distance”.

Corrected thank you.

Line 39: “a less” used here does not read well, perhaps “lower” or “reduced” would be better in its place.

Corrected thank you.

 Introduction

I think that the introduction is well written but can be hard to follow and would benefit from some restructuring and reworking to be more specific, and to flow better. The additional information that I suggest including below will help to make it more clear in detailing: the value of urban green spaces to natives, the need to manage them owing to introduced predator presence, how this is particularly apparent in NZ, how understanding microhabitat use of introduced predatory mammals could help with management applications of green spaces – with justification for your study, and the study details including research questions and brief methods and with some relevance to conservation significance and how the results may be used.

We have re-structured the introduction to emphasise the value of green spaces (1st paragraph), particular relevance to NZ (2nd paragraph), justification and how management might benefit from the information (last paragraph).

Line 47: there is a misplaced open parenthesis “(“at the start of this line.

Corrected, thank you.

Line 48: needs a comma after “green infrastructure”.

Comma added, thank you.

Line 46-49: this is a long and fragmented sentence that would read better if it were two and did not start with “while”.

We split the sentence into the following two: “Urban green spaces are most frequently the sites that provide such habitat [4,7,6,8]. The variety of green space types, from remnant patches of native vegetation, to highly artificial and engineered green infrastructure, such as green roofs [9,10], results in considerable variation across green spaces in terms of the biodiversity they support [11].

Line 52: would read better as: … urban green spaces provide,…

Amended as suggested.

Line 54: “(“ should be “[“.

Corrected, thank you.

Paragraph 2: I think it would be good here to state that NZ does not host any native mammalian predators, and to list the specific introduced predators here and how long they have been present and give some context to the extent of damage they can cause on native NZ species, before going on to discuss their utilisation of urban green spaces and how they sustain their populations. This new information is important to give context to the study, but only needs to be concise. Perhaps the information from lines 106-117 could be used around here, with the inclusion of the above mentioned information, to flow better into an argument for your research.

We have added some sentences into this paragraph stating that NZ does not host any native mammalian predators and listing the predators that we do have: “When humans first arrived, NZ ecosystems contained no land mammals other than some species of bat [2]. Currently there are 12 predatory mammals including four species of rodent (Rattus spp. & Mus musculus), three mustelids, European hedgehogs (Erinaceus europaeus), common brushtail possums (Trichosurus vulpecula), pigs (Sus scrofa), cats (Felis catus) and dogs (Canis lupus familiaris). Of these, all but pigs, Rattus exulans, and mustelids are commonly found in urban areas. Urban green spaces provide habitat that supports populations of these invasive predators.”

However, given that there are 12 species of mammalian predators, we would prefer not to provide a description for each of how long they have been in NZ and the extent of the damage they do as this would substantially lengthen the Introduction, even if the description for each was brief. We have brought some of the information provided in lines 106-117 up, but have left most of it where it is, as it is specific to the suite of species our study focuses on; if this information was brought up to the 2nd paragraph, we feel it would distract from the more general context we provide in the first few paragraphs.

Line 82: it is worth noting here, or somewhere in the introduction, that urban green spaces that have a level of vegetation complexity, be it native or not (there is support for both published), and connectivity, can support a diversity of native fauna that is often abundant despite the small habitat size and additional stressors of human disturbances and introduced predators. I know that there is a brief broad mention of this in the first paragraph and more in the last paragraph, but I think it could be more specific and should all be placed at the start of the introduction. I think it is important to detail more how urban green spaces can be used by both native and introduced species, and therefore play a valuable role in sustaining biodiversity. This also requires a level of consideration in the pest management of urban areas, to select methods that will not affect natives, and that the people in these areas will accept. This may be more apparent in countries other than NZ but as this is a global journal this needs to be acknowledged somewhere in the introduction too. It does not need to take up too much word count but could perhaps be achieved through changing and slightly extending what is already there.

We have re-written the first paragraph of the Introduction to incorporate these points as follows: “Despite large-scale habitat loss brought about by urban expansion [1,2], cities still provide habitat for many species, both native and introduced [3,4,5], and even refuges for some endangered species [6]. Urban green spaces are most frequently the sites that provide such habitat [4,7,6,8]. The variety of green space types, from remnant patches of native vegetation, to highly artificial and engineered green infrastructure, such as green roofs [9,10], results in considerable variation across green spaces in terms of the biodiversity they support [11]. Size, connectivity, vegetation composition and structure of green spaces all affect their capacity to sustain biodiversity [12-14]. Nevertheless, despite the often small size of habitat patches and the additional stressors of human disturbances and introduced predators, green spaces can support a diversity of native and introduced fauna, and play a valuable role in sustaining biodiversity [6,14].

We do address the social acceptability of predator control methods to some extent in the final section of the Discussion. We would prefer not to introduce this topic in the Introduction, as the focus of the study is on habitat associations, and acceptability of control measures is of course important, but not the focus of this study.

Line 84: as some of the species listed can be native in some countries, I think it would be good to continue to state “introduced mammals” each time, despite having stated it at the start. The suggested additional information for Paragraph 2 would also help to make this clearer.

Amended as suggested.

Line 92: is a bit general and it would be good if it could be re-written, or had more included, to better support the justification of this study, particularly after the results of similar are given.

This has been re-written to justify the study as follows: “Broad-scale, multi-habitat and multi-species research which allows us to understand factors influencing distributions of urban invasive predatory species across a range of urban green space types and in different urban centres should better inform efforts to limit negative impacts on native biodiversity.”

Line 96: the study did not aim, you did, please reword to either: “In this study we aimed to” or “The aim of this study was to, therefore,”.

Amended as suggested.

Line 101: I think if using common names then “black rat” is more globally used than “ship rats”, and if changing to this then “brown rat” is as widely used as “Norway rat” and provides an easy distinction between the two Rattus species.

Amended as suggested.

Line 102: as mustelids have not been previously mentioned, it is a bit confusing to mention them here. The suggestion made regarding Paragraph 2 should help clarify this.

We have introduced mustelids in paragraph 2 where we also explain that they are not commonly found in urban areas.

Further, this point needs some clarification. What exactly are “reasonable numbers”, please be more specific. Are you trying to state that you selected the “non-companion” introduced predators to study as they can be more readily managed via target trapping, unlike the introduced mustelids and domestic cats that are also present in the areas, though in smaller numbers, but are harder to manage owing to their relationships with people (as pets) and their aversions to traps? However, if this is what you are trying to state it is a bit misleading as mustelids and cats are mentioned in the methods and results.

We have re-written this paragraph as follows, which should clarify why mustelids and cats are mentioned in the results, and why we have excluded mustelids and cats and dogs from the analyses of habitat preferences. We determine the fine-scale and landscape-scale habitat characteristics that influence the distributions of the following five non-companion, mammalian, invasive predators found in urban green spaces: black rats, brown rats, brushtail possums, hedgehogs and house mice. These species are known to occur in all NZ cities and can be controlled using widely applied methods (e.g. traps). A better understanding of the distribution of these species across the city and their fine-scale habitat associations will allow more efficient control, such as more targeted trap placements. We do not examine the habitat preferences of mustelids, which were detected infrequently and are known to be uncommon in cities [31], nor of dogs and cats, as their companion animal status renders their control more complex [76], despite, in the case of cats, being important urban predators [77]. However, we do report on the prevalence of these species.

Line 104: perhaps “prevalent” is a better word than “important” here, but as stated above I think that these two sentences could do with a rework to make your point and justification for the species monitored clearer.

These sentences have been re-worked, but we have retained “important” which we feel is appropriate, as domestic cats are the most abundant predators in any NZ city.

The third last paragraph (96-105): as the aim of the study is first introduced here it would be good to follow on here with the research questions and how they will be briefly investigated, any predictions, then the conservation relevance that the results may have (briefly). This should be the concluding paragraph of the introduction. This will require removing the objectives listed in the below two paragraphs from the surrounding information, that as mentioned above and below would be much better placed earlier on in the Introduction, and placing all of the aims/objectives together, with some more detail to better clarify just what you are testing and how.

The aims of the study and conservation relevance are now all provided in the final paragraph. The information about species has been moved earlier in the introduction as suggested.

Line 118: This sentence would work better if flipped and made specific to the study, so that it read: We aimed to identify common features determining urban predator distributions in three cities (……) and three types of urban green space (……..). It would also be better placed with the preceding paragraph that details the aims of the study.

Sentence has been flipped and integrated into the previous paragraph.

The last two paragraphs (106-135): as mentioned above the details given in these paragraphs would be better placed earlier on in the introduction, before the details of your study, this will help build a better flow for the need of this study.

We have moved this information earlier in the introduction as suggested.

As the preceding paragraph (third from last) details the aims of the study this would be better to extend (see above) and close the introduction.

Aims etc are all now in the final paragraph.

 Materials and Methods

Line 140: what exactly is primary and secondary predominantly native forest?

We believe primary and secondary are common terms when applied to forest, but we have included “old growth” to further clarify primary forest.

Line 142: I think it would be good given the global audience to include here that amenity parks were “highly modified” and to also include at the end of the sentence if the vegetation patches were remnant native or introduced or both.

The sentence has been amended to: Amenity parks were highly modified areas (parks/reserves/sports fields), which were fringed with both native and introduced scattered trees, shrubs, and/or long grass.

Line 143: as in line 141 you state that “residential gardens” are a green space type that was sampled I think that it is best to continue with this labelling, and to replace “Household properties” at line 143 with “residential gardens”.

Amended as suggested.

Line 159: were the tracking tunnels baited with a food lure or not? Okay, I can see this is included at lines 181-184 and would be best to be moved up to here.

We have added some words to the first sentence to indicate that the tunnels were baited with a food lure, however we have left the more detailed information below, as it relates to both the tunnels and the cameras, so needs to follow the information about cameras.

Line 160: were the cameras infrared allowing for less disruptive night vision? If so, please insert “infrared” between motion-activated and cameras.

Added “infra-red”.

Statistical analyses: this section would benefit from an edit, there is much detail given on how to interpret the results, which could be really refined, and the detail that is given on the parameters of the models and how they were selected then ran is spread throughout and would be better placed together.

We would prefer to retain information on how results should be interpreted because many readers are unfamiliar with Bayesian statistics and our explanations ensure clear interpretation. We have removed some detail though.

We have reduced and streamlined the section on parameter selection.

The proceeding sections on the models provide some good information but again this is all very long and should be reduced into one single section on statistical analyses. As these sections are now, they appear disjunct with cut and paste text book style information in parts that could be presented more succinctly and relative to the study, which would display a better understanding of the processes used.

These sections have been collapsed into one section: Statistical analyses.

Lines 218-220: I am a bit concerned that the habitat surveys were taken based on categories and then the data was transformed for analyses. Would it not have been more accurate to just leave these as the direct percent cover for each when measuring, instead of grouping into categories that are not even and then transforming them back to a mid-percent value that is assessed as a proportion, thus giving two transformations (category, and mid-percent proportion) on the actual value which may give different results to what is naturally occurring, especially at the finer scale. I think that analyses would be better ran on the raw data, then based on the findings could be related to categories if they were evident.

We have re-written the paragraph explaining why we adopted this approach as follows: “Cover scores were converted to the mid-point percent cover of each category range i.e., RECCE cover scores 0–6 were converted to 0%, 0.5%, 3%, 15.5%, 38%, 63% and 88% cover, and in the residential garden models scores of 1–6 were converted to 5%, 17.5%, 38%, 63%, 83%, and 95% cover. Data used in the model were expressed as proportions. This was to allow for more intuitive and interpretable linear results, to reduce model instability caused by the excessive numbers of categories and allow for simpler aggregation of cover scores from multiple variables.”

Line 228: this could be reworded to be a bit clearer. For example, to something like: To test for the effects of fine- and landscape-scale green space habitat components on the presence of introduced predators in NZ we fit separate linear models for rats (grouping brown and black rats together), mice, hedgehogs, and possums using the program <insert name and reference here> and the package <(or setting) insert name and reference here>. Linear hierarchical models, using a Baysian framework, were created to……

We have re-written the section on statistical analysis.

Line 264: I think the word “rationale” could be inserted here before variable selection, to reduce confusion as to whether it was a step wise statistical assessment of the relevance of each variable that was used in selection processes.

We have amended this heading to  “Process of variable selection and model building” which we believe better reflects the content of this section.

 Results

This section was well presented and thorough, if anything despite the number of figures and tables given, I think that a summary visual representation of the results for each species relative to the finer-scale habitat parameters could be given together in one table or figure for easier quick reference than reading through the text, which is still adequate. However, I do wonder if the results would be different if modelled off the raw (uncategorised and unadjusted) data, as suggested above.

Given that there are already quite a few tables (9) and figures (4), and also given the variability across species in terms of their habitat associations at different scales, we feel another table/figure summarising everything would add one more table to those already presented, and wouldn’t necessarily be a simple representation of the results.

 Discussion

The Discussion section does well to support the results, but some language changes could be made to appear less bias. The Discussion section is overall very long and could do with some restructuring and refining, particularly the first section. It would flow better if each paragraph focused on an animal relative to the habitat components, instead of jumping back and forth between the two.

We have re-written a large part of the Discussion in sections focusing on each animal as suggested, and this has reduced the length of the manuscript.

Sections do appear to be grouped somewhat by habitat components or animal but are often confusing in their delivery, and repetitive of each other for an animal, which could be reduced if presented per animal per paragraph – as this is how it was modelled. These changes could also help in reducing the word count.

Thank you for the suggestion - we believe that our re-write has reduced repetition.

Line 561: there is a double space before the sentence starting with “Hedgehog detections”.

Corrected, thank you.

Line 565: from my experience brushtail possums in Australia are often present in higher densities in green space habitats that are closer to urban build-up than in the larger surrounding forest fragments, so I do not think that it is right to state “unsurprisingly” as it is really location and habitat dependent. There certainly are amble food and shelter resources in urban yards and the surrounds.

“Unsurprisingly” deleted.

Line 586-587: I think this needs to be more speculative than certain unless you can provide evidence to support it. For example, you could instead state: This may be relative to the likelihood of mice and possums being predated upon by domestic pets, and/or traps by people (and give references).

Amended as suggested.

Line 593-596: it is unclear here in which animal you are referring to, or if none of them found a preference for residential gardens. Further, the run-on of this sentence is confusing. Why is it then related to amenity parks? This should be two separate sentences. As it is written now it is not very clear why you are linking these two points together.

Thank you for pointing out this section which was confusing. We have amended the text to the following: “In New Zealand, hedgehogs do not always show strong preferences between urban habitat types [31]. In our study we did not find a preference for residential gardens, which typically have a higher density of structures. Hedgehogs were less likely to be detected in forests than in amenity parks, however our detection devices at amenity sites were placed in vegetation edging these playing fields, which may provide more resources for hedgehogs.

Line 646: there is a double space before the sentence starting with “The shrub layer”.

Corrected, thank you.

Section 4.3: I think it should also be noted that differences could be relative to other factors that were more influential for these areas that were not measured in this study. This highlights the importance for the need to clarify throughout the discussion that the results were relative to that given location under the observed conditions and may only be suggestive of what could occur at other locations.

  Good point. We have added a sentence towards the end of the first paragraph in the discussion to this effects, and  amended the text in this section towards the end of the discussion to read: “These results show large sample sizes are needed for this type of analysis and to accommodate high heterogeneity between cities, and that differences could be related to other factors that were more influential in these two cities but not measured in this study.

Reviewer 2 Report

Dear authors, congratulations for your work that I have found correct and precise in all its parts.

For this I suggest acceptance in presence form.

Best,

the reviwer

Author Response

Reviewer 2

Comments and Suggestions for Authors

Dear authors, congratulations for your work that I have found correct and precise in all its parts.

For this I suggest acceptance in presence form.

Thank you.

Reviewer 3 Report

The manuscript reports the presences of five invasive urban mammals in three urban green space types across three New Zealand cities, and across two seasons, and analyzed the influences of landscape and fine-scale factors on the presences of the mammals in urban green land. The results are very interesting to urban biologists, designers and managers. Because there are too many kinds of mammals, cities and factors considered in the MS, the results are showed on triviality, so that the key findings in the MS are not clearly focused on. I would like to give following suggestions for improving the MS:

(1)    In the data analysis, CI of odds ratio can confirm if the probability of the presence of a mammal is significant at each situation. However, the more interesting issue is if the differences between cities, green space types, or other factors is significant. It is important to analyze the influences of each kind of factors separately. And the results are organized by influencing factors.

(2)    In the discussion and conclusion, the similarity in distribution pattern and factors for all 5 mammals should be derived. The MS should be shortened for clarity and main finding.

(3)    In result section, lists of numbers for OR, CI, and MPE is too many that is not good for reading experience.

Author Response

Reviewer 3

Comments and Suggestions for Authors

The manuscript reports the presences of five invasive urban mammals in three urban green space types across three New Zealand cities, and across two seasons, and analyzed the influences of landscape and fine-scale factors on the presences of the mammals in urban green land. The results are very interesting to urban biologists, designers and managers.

Thank you.

Because there are too many kinds of mammals, cities and factors considered in the MS, the results are showed on triviality, so that the key findings in the MS are not clearly focused on. I would like to give following suggestions for improving the MS:

(1)    In the data analysis, CI of odds ratio can confirm if the probability of the presence of a mammal is significant at each situation. However, the more interesting issue is if the differences between cities, green space types, or other factors is significant. It is important to analyze the influences of each kind of factors separately. And the results are organized by influencing factors.

Here the reviewer appears to be suggesting we focus only on the larger scale factor level differences in detection by city, season etc. We do report on the significance of the detection odds for each species between cities (section 3.5) and show that, for example, probability of rat presence was higher in Wellington than in Dunedin or Hamilton but we also report on difference between green space types. However, a major focus of our study is the fine-scale habitat characteristics associated with predator presence, as it is this kind of information that will inform the design of predator control operations.

(2)    In the discussion and conclusion, the similarity in distribution pattern and factors for all 5 mammals should be derived. The MS should be shortened for clarity and main finding.

The first paragraph of the discussion focuses on general patterns across cities - both similarities and differences. We have re-written a large part of the discussion into sections focusing on each species, which has reduced repetition and we hope improved clarity.

(3)    In result section, lists of numbers for OR, CI, and MPE is too many that is not good for reading experience.

These are all key components to the results. The reviewer may be referring to the section where factor levels are compared/interactions effects presented e.g. “detection odds in Dunedin were significantly higher than in Hamilton (OR = 71.52, CI = 21.2 – 275, MPE = 1)”. We agree that there are a lot of results present here, but the only other option is to create two big tables with all the different comparisons and associated numbers listed - these would cover two pages, and we feel the way we have presented the figures is the more palatable of the options.

Round 2

Reviewer 3 Report

The authors have responded correctly to my review . So I recommand to accept it at the present form. 

Author Response

Thank you for your comments.